# PFN1 Inhibits Myogenesis of Bovine Myoblast Cells via Cdc42-PAK/JNK

**DOI:** 10.3390/cells11203188

**Published:** 2022-10-11

**Authors:** Jingjing Zi, Jing Xu, Jintang Luo, Xu Yang, Zhen Zhen, Xin Li, Debao Hu, Yiwen Guo, Hong Guo, Xiangbin Ding, Linlin Zhang

**Affiliations:** Tianjin Key Laboratory of Agricultural Animal Breeding and Healthy Husbandry, College of Animal Science and Veterinary Medicine, Tianjin Agricultural University, Tianjin 300392, China

**Keywords:** bovine, skeletal muscle satellite cells, myogenic differentiation, PFN1, Cdc42, PAK, JNK

## Abstract

Myoblast differentiation is essential for the formation of skeletal muscle myofibers. Profilin1 (Pfn1) has been identified as an actin-associated protein, and has been shown to be critically important to cellular function. Our previous study found that PFN1 may inhibit the differentiation of bovine skeletal muscle satellite cells, but the underlying mechanism is not known. Here, we confirmed that PFN1 negatively regulated the myogenic differentiation of bovine skeletal muscle satellite cells. Immunoprecipitation assay combined with mass spectrometry showed that Cdc42 was a binding protein of PFN1. Cdc42 could be activated by PFN1 and could inhibit the myogenic differentiation like PFN1. Mechanistically, activated Cdc42 increased the phosphorylation level of p2l-activated kinase (PAK), which further activated the phosphorylation activity of c-Jun N-terminal kinase (JNK), whereas PAK and JNK are inhibitors of myogenic differentiation. Taken together, our results reveal that PFN1 is a repressor of bovine myogenic differentiation, and provide the regulatory mechanism.

## 1. Introduction

Satellite cells are a type of specific adult stem cell population of skeletal muscle, which are normally quiescent. However, satellite cells can be strongly activated after tissue injury, and begin to divide and proliferate into myoblasts. These myoblasts then fuse and differentiate into muscle fibers, thus supporting skeletal muscle regeneration [1]. Myoblast differentiation is a complex process, which is regulated by many genes. Myogenin (MyoG) and Myosin heavy chain (MyHC) are the markers of myoblast differentiation, which are highly expressed in differentiated myoblasts, and can promote cell fusion and form myotubes in vitro [2,3]. In the previous study, we confirmed that a lncRNA-lnc23 which promoted myoblast differentiation could bind to the PFN1 protein, and preliminarily confirmed that interfering with PFN1 promoted the differentiation of bovine myoblasts [4], which suggests that PFN1 has a certain regulatory effect on the differentiation of bovine myoblasts. However, the specific mechanism of the effect of PFN1 on the differentiation of bovine myoblasts has not been well elucidated, so this study continues to study the specific role of PFN1 in the process of myoblast differentiation, in an attempt to reveal the specific mechanism of PFN1 regulation of myoblast differentiation, and provide new data for understanding the complex gene expression regulatory network during bovine muscle development.

PFN1 is one of the important actin-binding proteins, the molecular weight of which is about 15 KDa. The tertiary structure of PFN1 is mainly composed of four α helixes and seven β folds [5]. PFN1 regulates the aggregation of actin cytoskeleton by combining with actin to form the PFN1–actin complex, which has been proved to play a key role in cell migration, cell division, neuronal differentiation, and synaptic plasticity. Mouse embryos without PFN1 cannot survive at the two-cell stages [6]. Mutations or deletions of PFN1 can also lead to some human diseases, which have become a hot topic of research. For example, Zou et al. found that PFN1 could inhibit the proliferation of breast cancer cells, and the expression of PFN1 could up-regulate the *PTEN* gene, thus inhibiting the AKT pathway, resulting in the increase of p27 protein [7,8]. The binding of p27 and Cdk2 complex makes the cell cycle stagnant in G1 phase, and finally inhibits cell proliferation [7,8]. Another study found that PFN1 changed the dynamics of the cytoskeleton by activating JAK2-STAT3, JNK, and P38-MAPK pathways, which promoted the thickening of vascular wall and the decrease of vascular lumen diameter, and finally led to hypertension [9]. Some studies have also shown that PFN1 mutations can lead to amyotrophic lateral sclerosis (amyotrophic lateral sclerosis, ALS) through different mechanisms [10,11,12]. Wu et al. found that wild-type PFN1 was dispersed in the cytoplasm, but the mutant PFN1 (C71G,G118V, and M114T) existed as a cytoplasmic ubiquitin aggregate. One of the markers of ALS is the formation of aggregates, which further indicates that PFN1 mutation induces protein aggregation and leads to ALS [13]. Therefore, PFN1 has been known to play an important role in cancer and human diseases, but there are few studies on bovine muscle growth and development. Our previous studies have preliminarily confirmed that interfering with PFN1 can promote the myogenic differentiation of bovine myoblasts, but the specific molecular mechanism is not clear. In this study, we will further study the effect of PFN1 on bovine myoblast differentiation and analyze the molecular mechanism.

The small GTPase protein, Cdc42, is a member of Rho GTPase and is widely present in cells. Cdc42 can regulate many physiological processes, such as cell proliferation, movement, polarity, cell division, and growth [14]. In mouse embryos with skeletal muscle conditional knockout of Cdc42 or Rac1, the protein levels of ARP2/3 complex and VASP and the number of F-actin were significantly lower than those of normal mice, the fusion index of muscle satellite cells was significantly lower than that of normal mice, and the muscle fibers formed were thinner and more disordered than those in normal mice [15]. Some previous studies suggested Cdc42 could regulate myogenesis by activating JNK, PAK, or the p38 MAPK in rat or mouse cells [16,17,18]. All these show that Cdc42 plays an important role in the growth and development of muscle. However, the functions of CdC42 in bovine skeletal muscle development are entirely unknown.

In this study, we find that PFN1 negatively regulates the differentiation of bovine skeletal muscle cells. Cdc42 is identified as a binding protein of PFN1, and has a negative regulatory effect on the differentiation of skeletal muscle cells consistent with PFN1. Mechanistically, PFN1 interacts with Cdc42 to stimulate Cdc42 activity, thereby activating PAK and JNK signaling pathways to inhibit bovine myoblastic differentiation.

## 2. Materials and Methods

### 2.1. Cell Isolation and Culture

The bovine myoblasts used in this study were prepared in our lab; the isolation, identification, and induced differentiation procedure were described in our previous study [19]. Myoblasts were recovered and resuspended in growth medium, Dulbecco’s modified Eagle’s medium (DMEM/High Glucose, Hyclone, Logan, UT, USA) with 20% fetal bovine serum (FBS, Gibco, Waltham, MA, USA), and 1% penicillin-streptomycin liquid (Solarbio, Beijing, China). The cell was cultured in a 37 °C, 5% CO_2_ incubator until the cell density reached 80%, and the proliferating cells were collected, or the differentiation medium containing DMEM with 2% horse serum (HS, Gibco, Waltham, MA, USA) and 1% penicillin-streptomycin liquid was aspirated to induce the cells to differentiate. After culturing in cell differentiation medium for 2 days (DM2), experiments were usually carried out.

### 2.2. Antibody and Inhibitors

The primary antibodies used in the study included rabbit anti-Profilin1 (1:1000, Abcam, ab50667), rabbit anti-Cdc42 (1:1000, Proteintech, Rosemont, IL, USA), rabbit anti-PAK (1:1000, Proteintech, Rosemont, IL, USA), rabbit anti-Phospho PAK (1:1500, Proteintech, IL, USA), rabbit anti-IgG (1:50, Santa Cruz, Dallas, TX, USA), rabbit anti-JNK (1:1000, Abcam, ab179461, London, UK), rabbit anti-Phospho JNK(1:1000, Abcam, ab124956, London, UK), mouse anti-MyHC (1:100, DSHB, CO, USA), mouse anti-MyoG (1:100, DSHB, Iowa, IA, USA), mouse anti-GAPDH (1:1000, Zhongshan, Golden Bridge Bio-technology, Beijing, China), and mouse anti-tubulin (1:3000, Proteintech, Rosemont, IL, USA). The secondary antibodies included horseradish peroxidase goat anti-mouse/rabbit IgG (1:10000, Zhongshan, Golden Bridge Bio-technology, Beijing, China), PAK inhibitors (Selleck, PF-3758309), and JNK inhibitors (GLPBIO, SP600125). A G-LISA^®^ Cdc42 Activation Assay Biochem Kit™ (cytoskeleton, Cat. # BK127-S) was used.

### 2.3. qRT-PCR Analysis

The total RNA of cells were isolated using the Cell RNA Extraction Kit (Aidlab, Beijing, China). Then, the first strand cDNA was prepared using PrimeScript II 1st Strand cDNA Synthesis Kit (Takara, Dalian, China). qRT-PCR was performed using All-in-One™ qRT-PCR Mix (Genocopoeia, Guangzhou, Chian) in a LightCycler^®^ 96 Instrument (Roche, Munich, Germany). GAPDH was used as an internal reference gene. The relative expression level of genes was calculated by the 2^−ΔΔCt^ method. The sequences of all primers used were listed as follows: GAPDH forward: 5′-TGTTGTGGATCTGACCTGCC-3′, reverse: 5′-AAGTCGCAGGAGACAACCTG-3′; MyoG forward: 5′-GGCTGACAAATGCCAGACTATCC-3′, reverse: 5′-TGGTCCCTTGCTTTATCTCCCT-3′; MyHC forward: 5′-CTGGAATCCGGAGGCAGAA-3′, reverse: 5′-TTTTCGAAGGTAGGGAGCGG-3′; Cdc42 forward: 5′-AAACAGCTGCCCCTACTGTC-3′, reverse: 5′-GGTTTGTTAGGGCTGCCTGA-3′; PFN1 forward: 5′-GTGGAGCCCCAACCTTCAAT-3′, reverse: 5′-TTGATCATACCGCCGTGGAC-3′.

### 2.4. Western Blot Analysis

Bovine myoblasts were washed by PBS buffer and lysed in RIPA buffer (Solarbio, Beijing, China) supplemented with PMSF for total protein characterization. Then, equal amounts of cells lysate were resolved by 10% or SDS-PAGE and transferred onto PVDF membranes (Millipore, Burlington, MA, USA). The membranes were blocked with 5% BSA for 1h, incubated with primary antibody at 4 °C for 2 h or overnight, then incubated with secondary antibody for 1h to detection using ECL chemiluminescent substrate (Solarbio). Quantitative analysis was carried out by calculating the gray value, which was calculated using ImageJ software (Bio-rad, Hercules, CA, USA) and the value of the target protein was normalized to internal reference protein.

### 2.5. Plasmids, siRNAs and Transfection

We obtained the recombinant plasmids pcDNA-PFN1 and pcDNA-Cdc42 by ligating the CDS regions of *PFN1* and Cdc42 to the pcDNA3.1(+) vector, respectively. The pcDNA3.1(+) was a negative control. All the primer sequences of gene cloning are listed in Appendix A. The siRNAs for *PFN1* and *Cdc42* knockdown and negative control sequences (si-NC) were synthesized by RiboBio (Guangzhou, China); the siRNA sequences are listed in Appendix A. The cells with 60–70% confluence in growth medium were transfected with plasmid or siRNA using Lipofectamine 3000 (Invitrogen, Waltham, MA, USA). Twelve hours after transfection, the cells were changed into a differentiation medium to induce cell differentiation.

### 2.6. Co-IP

Co-IP was performed using a Beaver BeadsTM Protein A/G Immunoprecipitation Kit (Beaver Biomedical Engineering Co., Ltd., Suzhou, China). Briefly, the protein of DM2 myoblasts were collected using binding buffer supplemented with 1× protease inhibitor cocktail. The pretreated magnetic beads protein A/G and the PFN1 antibody or IgG antibody were incubated for 30 min at room temperature. Then, the beads–antibody complex and cell lysates were incubated with rotating at 4 °C overnight. Finally, the supernatants were subjected to SDS-PAGE and mass spectrometry (MS) identification by Wuhan GeneCreate Biological Engineering Co., Ltd (Wuhan, China).

### 2.7. Cdc42 Activity Assay

A Cdc42 activity assay was performed using a G-LISA^®^ Cdc42 Activity Assay Biochem Kit (Beaver Biosciences, Guangzhou, China). Briefly, the cells were lysed with lysis buffer containing 1% protease inhibitor cocktail, and the diluted Cdc42 primary antibody was added and incubated for 15 min in the micro-vibrator. Then, the diluted Cdc42 secondary antibody was added and incubated for 30 min in a micro-vibrator. Finally, the mixed HRP A/B was added, incubated at 37 °C for 15 min, and the absorbance value of 490 nm was immediately measured.

### 2.8. Statistical Analysis

All of the experiments were performed at least three times. All results are presented as the mean ± SEM. Statistical analyses of differences between groups were performed using a two-tailed Student’s t-test, and *p* < 0.05 was considered statistically significant; * *p* < 0.05, ** *p* < 0.01, and *** *p* < 0.001.

## 3. Results

### 3.1. PFN1 Inhibits Bovine Myoblasts Differentiation

To examine the expression of PFN1 during myoblast differentiation, the mRNA and protein expression levels of PFN1 at different differentiation days were assessed. The results showed that the expression of PFN1 decreased significantly during the myoblast differentiation. To determine the function of PFN1 in this process, we transferred PFN1 siRNA or pcDNA-PFN1 into the bovine myoblasts to gain the knockdown or overexpress of PFN1 cells, respectively. Myoblast differentiation marker genes, MyHC and MyoG, were detected by qRT-PCR and Western blot in DM2. We found a significant increase both at the mRNA and protein levels of MyHC and MyoG in PFN1 knockdown cells, whereas the mRNA and protein levels of MyHC and MyoG significantly decreased in PFN1 overexpression cells (Figure 1A–D). Moreover, correspondingly, the number and diameter of myotubes in the knockdown cells were increased compared to control cells (Figure 1E). On the contrary, the overexpression cells formed less and thinner myotubes compared to the control cells (Figure 1F). Taken together, these results demonstrated that PFN1 inhibited the differentiation of bovine skeletal muscle satellite cells.

### 3.2. Analysis of PFN1 Potentially Binding Proteins

According to previous research, PFN1 usually plays a regulatory role by interacting with other proteins. Thus, we attempted to identify the interacting protein of PFN1 to analyze the regulation mechanism of PFN1 on myogenic differentiation. We successfully obtained the binding protein of PFN1 in DM2 myoblasts by immunoprecipitation technology (Appendix A). Then, the binding protein of PFN1 was analyzed by mass spectrometry. A total of 634 and 13 proteins were, respectively, identified from IP and IgG, of which 624 proteins, specifically in the IP group, were deemed as PFN1 potential binding proteins (PBPs) (Figure 2A). Subsequently, we also performed GO and KEGG analysis of PFN1 PBPs. The results exposed that the PBPs were remarkably enriched in the tricarboxylic acid cycle, signal transcription mediated by small GTPase, actin filament assembly, myocardial contraction, muscle contraction, and so on, in the biological process (Figure 2B). In the cellular component, these proteins mainly belonged to the mitochondrion, cytosol, melanosome, actin filament, and so on (Figure 2C). Additionally, molecular function mainly includes actin filament binding, GTP binding, GTPase activity, etc. (Figure 2D). KEGG enrichment analysis showed the PBPs were significantly enriched in carbon metabolism, fatty acid degradation, metabolic pathway, and glycolysis/gluconeogenesis and proteoglycans in cancer (Figure 2E).

These pathways are also related to muscle development, which further proves the key regulatory role of PFN1 in myogenic differentiation. With the aid of the previous bioinformatics analysis, it was found that the differential binding proteins were mostly related to actin and small GTPase, and these proteins were also related to muscle development. Protein–protein interaction (PPI) analysis of PBPs by the STRING database revealed that PFN1 interacted with Cdc42, RhoA, RhoC, VASP, and so on (Figure 2F). After comprehensive consideration, Cdc42, a member of the small GTPase family, was selected as the candidate binding protein of PFN1 for further study.

Further, in order to validate the binding of PFN1 and Cdc42, proteins of DM2 myoblasts were collected and subjected to CoIP. The IP efficiency was detected by Western blot. The results showed that Cdc42 was significantly enriched in the PFN1-IP group compared with negative IgG (Figure 2G). Thus, Cdc42 was further identified as the interacting protein of PFN1.

### 3.3. PFN1 Inhibits the Differentiation of Bovine Myoblasts via Cdc42

Next, to illuminate the mechanism of PFN1 cooperating with Cdc42 in regulating the cell differentiation, the effects of PFN1 on Cdc42 were measured both at mRNA and protein levels. The results showed that changing the expression of PFN1 did not significantly effect the mRNA level of Cdc42 in DM2 (Figure 3A); however, after down-regulating or up-regulating PFN1 expression, the protein level of Cdc42 significantly decreased or increased, correspondingly (Figure 3B,C). Hence, we speculated that PFN1 might affect the amount of Cdc42 protein by regulating translation or stability rather than transcription. Furthermore, in view of the important regulatory role of Cdc42 activity on cell proliferation and differentiation, we detected the effect of PFN1 on Cdc42 activity in DM2 myoblasts. The results showed that after down-regulation or up-regulation of PFN1, respectively, the activity of Cdc42 significantly decreased or increased (Figure 3D). Cumulatively, these outcomes showed that PFN1 had a positive regulatory effect on Cdc42.

Considering that although it had been reported that Cdc42 could regulate myoblast differentiation in other species [15], the effect of Cdc42 on bovine myoblast differentiation had not been reported, we detected the effect of Cdc42 on bovine myoblast differentiation. Thus, to investigate whether Cdc42 impacted the cell differentiation, Cdc42 was interfered or overexpressed in bovine myoblasts, respectively, and we tested its effect on differentiation in DM2 myoblasts. The interference of Cdc42 resulted in significant elevated mRNA and protein levels of MyoG and MyHC (Figure 3E,G), and the formation of more pronounced myotubes in DM2 cells (Figure 3I). On the contrary, the overexpression of Cdc42 resulted in a significant decrease in the amount of mRNA and protein of MyoG and MyHC (Figure 3F,H), and the formation of less strong myotubes in the cells (Figure 3J). These results indicated that Cdc42 inhibited the differentiation of myoblasts, which was similar to PFN1.

Based on the above findings, we concluded that PFN1 activated Cdc42 activity by binding to Cdc42, consequently inhibiting the cell differentiation.

### 3.4. PFN1-Cdc42 Negatively Regulates the Differentiation of Myoblasts via Activating PAK/JNK Signaling Pathway

Based on above data, we speculated that PFN1 could negatively regulate the differentiation of bovine myoblasts by activating Cdc42. Uncovering the detailed regulation mechanism of PFN1-Cdc42 on inhibiting myogenic differentiation is needed. Some studies have reported that Cdc42 can activate the PAK signaling pathway to affect cell activity, and the PAK signaling pathway has been confirmed to play important roles in the differentiation of myoblasts. Thus, we verified whether Cdc42 regulates myogenic differentiation through the PAK signaling pathway in bovine myoblasts. The phosphorylation of PAK is a sign of activation of the PAK signaling pathway. We found that the knockdown of Cdc42 could decrease the phosphorylation levels of PAK in DM2 of the bovine myoblasts (Figure 4A), whereas overexpression of Cdc42 could increase the phosphorylation levels of PAK (Figure 4B). These results demonstrated that Cdc42 could indeed activate PAK in the bovine myoblasts, and we went on to explore the effect of PAK on myogenic differentiation. In order to achieve the ideal inhibitory effect without affecting the living of cells, we first explored the appropriate concentration of the PAK inhibitor, and chose 0.5 μM as the final experimental concentration (Appendix A). Under treatment of the PAK inhibitor, it was found that with the decrease of phosphorylation level of PAK, the protein content of MyoG and MyHC increased significantly (Figure 4C). This result indicated that p-PAK could inhibit the myogenic differentiation, considering that activated PAK can activate JNK, which has been reported to regulate myoblast differentiation. Therefore, we checked whether PAK had the same effect on JNK in bovine myoblasts. As expected, we found that PAK inhibitors significantly decreased the phosphorylation level of JNK (Figure 4D), indicating that PAK could positively regulate the activity of JNK in the bovine myoblasts as in other cells. Based on all the above results, we suspected that PFN1 interacted with Cdc42 to activate PAK, and the activated PAK further activated JNK, which could negatively regulate myogenic differentiation, thus inhibiting myogenic differentiation of the bovine myoblasts. If this is the case, PFN1 and Cdc42 should be able to indirectly regulate the activity of JNK. To confirm this conjecture, thus, we detected the effect of PFN1 and Cdc42 on the activity of JNK. Western blot showed that the protein levels of p-JNK were significantly reduced in PFN1 or Cdc42-knockdown cells and significantly increased in PFN1 or Cdc42-overexpressing cells (Figure 4E–H), which suggested, as we speculated, that the expression of PFN1 or Cdc42 did positively stimulate the activity of JNK. Furthermore, we confirmed that the protein expression of MyoG and MyHC was significantly up-regulated when the JNK phosphorylation level was inhibited in DM2 cells (Figure S3 and Figure 4I), which indicates that the JNK signal pathway plays a role in inhibiting myogenic differentiation in the bovine myoblasts.

Combining all the results, we considered that PFN1-Cdc42 could activate the PAK/JNK signaling pathway, thereby negatively regulating the differentiation of bovine myoblasts.

## 4. Discussion

PFN1 is an evolutionarily conserved small actin-binding protein that regulates many cellular activities, including proliferation and motility [20]. According to research findings, the overexpression of PFN1 significantly inhibits the proliferation, migration, and invasion of hepatocellular carcinoma cells [21]. Janke et al. found that compared with control cells, the expression level of PFN1 in tumorigenic breast cancer cells was reduced [22]. Zou et al. found that PFN1 was also involved in regulating the migration of breast cancer cells [23]. A gene-silencing study found that PFN1 was involved in the regulation of endothelial cell migration and proliferation [24]. In our previous study, we initially found that interfering with PFN1 could promote the differentiation of bovine myoblasts. In this study, we confirmed that PFN1 could indeed negatively regulate the differentiation of bovine myoblasts.

An increasing amount of research is showing that PFN1 can act as a key signaling molecule in the network hub, and participate in a series of important physiological processes, such as protein transmembrane transport, nuclear transcription, and activation of the small GTPase signaling pathway, by binding to target proteins through its domain [25]. Thus, we investigated the mechanism of PFN1 on regulating myogenic differentiation via analyzing proteins regulated by PFN1. Firstly, the binding proteins of PFN1 were captured by IP technology in bovine DM2 myoblasts, and the binding proteins were analyzed and identified by mass spectrometry. Then, we analyzed the PFN1 binding proteins by GO, KEGG, and STRING, and found a small GTPase-Cdc42 involved in muscle development. We chose Cdc42 as the binding target protein of PFN1, and confirmed the binding of PFN1 to Cdc42 by CoIP.

As a Rho GTPase, Cdc42 plays a key role in a variety of cellular processes, and activated Cdc42 can mediate signaling cascades and activate various downstream effectors, thereby leading to many cellular activities, such as cell adhesion, motility, polarity, cytokinesis, and growth [26,27,28]. In the late stage of skeletal muscle differentiation, the expression of active Cdc42 increases, which plays an important role in myogenic differentiation [29]. The effect of Cdc42 on myogenesis has been demonstrated in mouse, human, and avian cells [15,17,30,31], but whether it is positive or negative remains controversial. A previous study showed that myoblast fusion required Rac1- and Cdc42-dependent actin polymerization in mice [15]. Another study found a Bin3-dependent pathway is a major regulator of Rac1 and Cdc42 activity in differentiated muscle cells, and Bin3 regulated Rac1- and Cdc42-dependent processes in mice myogenesis [30]. Kang et al. found that the promyogenic cell surface receptor, Cdo, could associate with the Cdc42 binding protein, Bnip-2, and this interaction activated Cdc42, which, in turn, stimulated p38α/β activity and promoted cell differentiation [16]. These studies have shown that Cdc42 plays an important positive regulatory role in myogenic differentiation, but there are also some studies with the opposite view. An earlier work by Gallo et al. showed that constitutively active Rac1 and Cdc42 inhibited the expression of MyoG and blocked myogenic differentiation in avian myoblasts [31]. Wu et al. suggested that CEP2 (a member of the Cdc42 subfamily) regulates myogenic differentiation by acting as a downstream factor of Cdc42 [32]. It is worth noting that a dual role of Cdc42 for myogenesis was proposed: a positive role by activating the SRF and P38 pathways and a negative role by activating the JNK pathway, respectively [17]. The inhibition of myogenesis by Cdc42, correlating with JNK activation, is consistent with our current study. In our current study, we have solid evidence that in bovine myoblasts, Cdc42 is an effector of PFN1, and Cdc42 can negatively regulate myogenesis.

Active Rho GTPases can interact with a number of downstream effectors, of which more than 200 effectors have been identified, e.g., some protein kinases such as ROCK and PAK family members, and scaffold proteins such as mDia1 and WASP [33]. PAK has been shown to activate JNK, SRT, and NFKB [34,35,36,37]. Among these upstream and downstream factors, PAK and JNK have been confirmed to play important roles in the differentiation of myoblasts. It has been reported that, in the metastasis of the rhabdomyosarcoma (RMS) cell line, guanine nucleotide exchange factor T(GEFT) accelerates the tumorigenicity by activating Rac1/Cdc42-PAK signaling [38]. Cdc42 could activate PAK to make it phosphorylated, and activated PAK could phosphorylate myosin light chain kinase (MLCK), resulting in the decrease of MLCK activity and MLC phosphorylation level, and finally inhibiting cell spreading [18]. Moreover, phosphorylated PAK could also inactivate MyHC and inhibit actin stress fiber formation [39]. From these results, we speculate that Cdc42 may affect the differentiation of bovine myoblasts by regulating the phosphorylation level of PAK. Finally, we confirmed that Cdc42 could indeed activate the phosphorylation level of PAK, and inhibiting of phosphorylation of PAK could promote the expression level of the differentiation markers, MyoG and MyHC. The JNK signaling pathway is another subclass of the MAPK signaling pathway, which can regulate important cell activities, including cell proliferation, differentiation, and apoptosis [40]. Some studies have shown that PAK can activate JNK in other cells [40], but there is no similar study in myoblasts. However, we already know that Cdc42 can activate JNK and inhibit differentiation in the myoblasts of rats and mice [17,32]. Therefore, we predicted that Cdc42 may activate JNK by activating PAK, and then inhibit myogenic differentiation in bovine myoblasts. As predicted, our study confirmed that when the phosphorylation activity of PAK was inhibited, the activity of JNK was also inhibited. Moreover, we also confirmed that after the inhibition of JNK, the expression levels of the myogenic differentiation markers, MyoG and MyHC, increased, indicating that JNK could inhibit the myogenesis of bovine myoblasts, which was consistent with previous studies in other species [17,32].

## 5. Conclusions

In conclusion, we confirmed that PFN1 played an important negative role in the differentiation of bovine myoblasts. Cdc42 is identified as a binding protein of PFN1, and has a negative regulatory effect on the differentiation of myoblasts consistent with PFN1. Mechanistically, PFN1 interacts with Cdc42 to stimulate Cdc42 activity, thereby activating PAK and JNK signaling pathways to inhibit bovine myoblastic differentiation.

## Figures and Tables

**Figure 1 cells-11-03188-f001:**
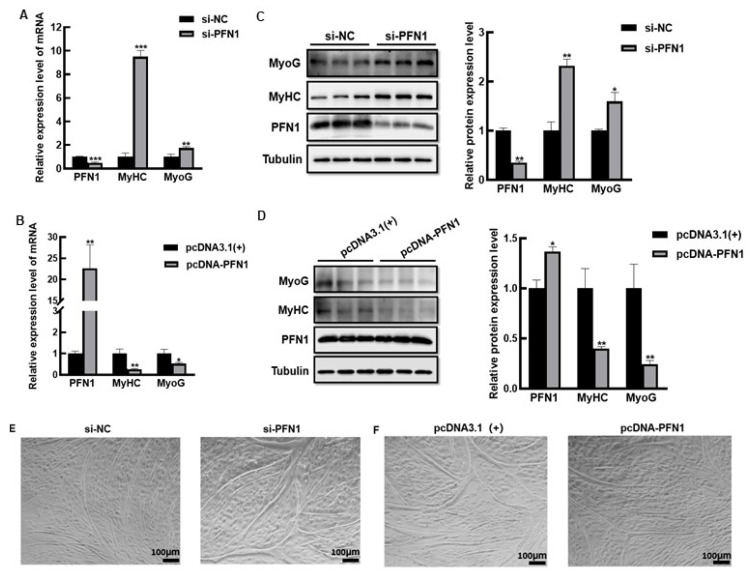
The effect of PFN1 on the differentiation of bovine skeletal muscle satellite cells: (**A**) qRT-PCR detection of mRNA expression level of differentiation marker genes in si-PFN1 or si-NC transfected DM2 myoblasts; (**B**) qRT-PCR detection of the mRNA expression level of differentiation marker genes in pcDNA-PFN1 or pcDNA3.1(+) transfected DM2 myoblasts; (**C**) Western blot and quantitative analysis of differentiation marker protein levels in si-PFN1 or si-NC transfected DM2 myoblasts; (**D**) Western blot and quantitative analysis of differentiation marker protein levels in pcDNA-PFN1 or pcDNA3.1(+) transfected DM2 myoblasts; (**E**) Myotube formation status in si-PFN1 or si-NC transfected DM2 myoblasts; (**F**) Myotube formation status during differentiation in pcDNA-PFN1 or pcDNA3.1(+) transfected DM2 myoblasts. si-NC stands for negative control, and pcDNA3.1(+) stands for empty vector control negative control. * *p* < 0.05, ** *p* < 0.01, and *** *p* < 0.001.

**Figure 2 cells-11-03188-f002:**
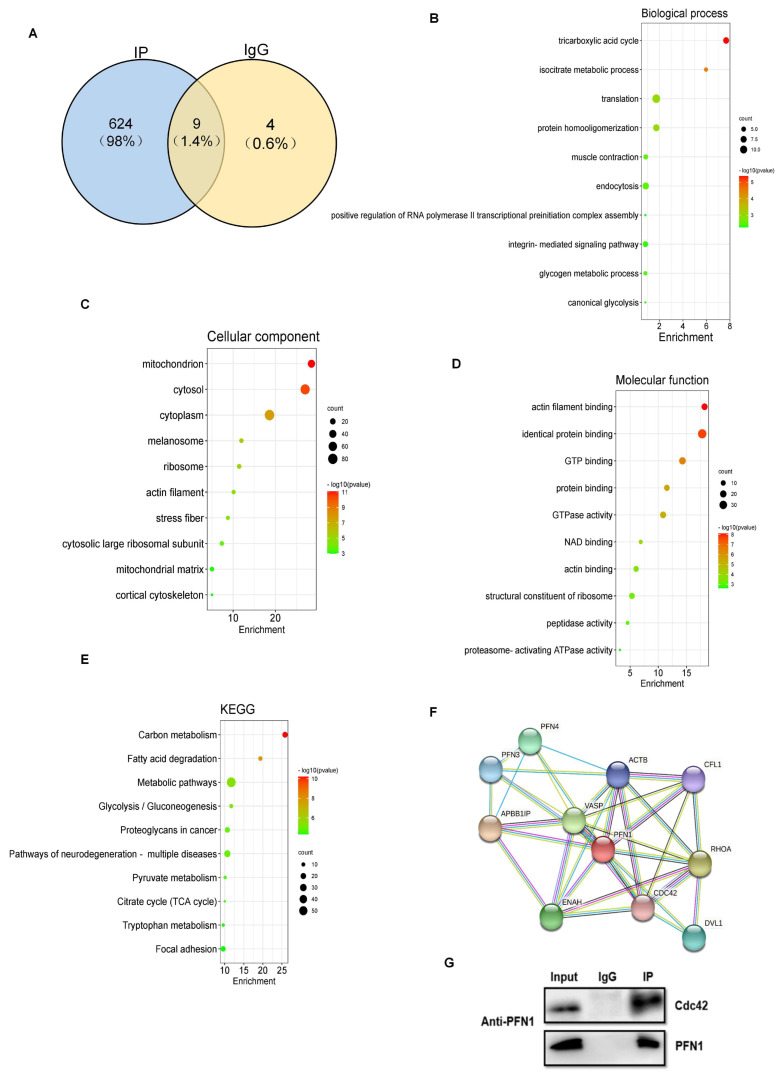
Identification of PFN1-interacting proteins: (**A**) identified proteins by CoIP-MS; (**B**–**E**) GO and KEGG analysis of PFN1 binding proteins; (**F**) protein-protein interaction analysis of PFN1 candidate binding proteins by STRING; (**G**) validation the binding of PFN1 and Cdc42 by Co-IP in DM2 myoblasts.

**Figure 3 cells-11-03188-f003:**
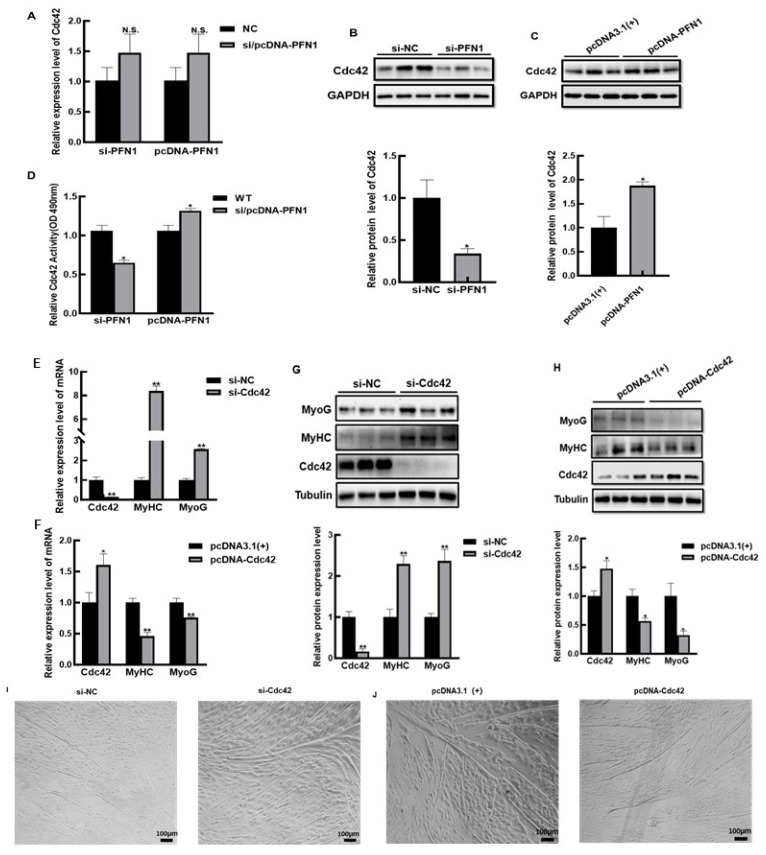
Regulation of Cdc42 by PFN1 and the effect of Cdc42 on the differentiation of bovine myoblasts: (**A**) qRT-PCR detecting the effect of interfering or overexpression of PFN1 on Cdc42 mRNA level in DM2 myoblasts; (**B**) Western blot detecting the effect of interfering with PFN1 on Cdc42 protein level in DM2 myoblasts; (**C**) Western blot detecting and quantitative analysis of the effect of overexpression of PFN1 on Cdc42 protein levels during differentiation; (**D**) the Cdc42 activity analysis in interfering or overexpression of PFN1 DM2 myoblast; (**E**) qRT-PCR detection of the mRNA expression level of differentiation marker genes in si-Cdc42 or si-NC transfected DM2 myoblasts; (**F**) qRT-PCR detection of the mRNA expression level of differentiation marker genes in pcDNA-Cdc42 or pcDNA3.1(+) transfected DM2 myoblasts; (**G**) Western blot analysis of differentiation marker protein levels in si-Cdc42 or si-NC transfected DM2 myoblasts; (**H**) Western blot detecting and quantitative analysis of differentiation marker protein levels in pcDNA-Cdc42 or pcDNA3.1(+) transfected DM2 myoblasts; (**I**) myotube formation status in si-Cdc42 or si-NC transfected DM2 myoblasts; (**J**) myotube formation status in pcDNA-Cdc42 or pcDNA3.1(+) transfected DM2 myoblasts. si-NC stands for negative control, and pcDNA3.1(+) stands for empty vector control negative control. N.S.: non significance, * *p* < 0.05, ** *p* < 0.01, and *** *p* < 0.001.

**Figure 4 cells-11-03188-f004:**
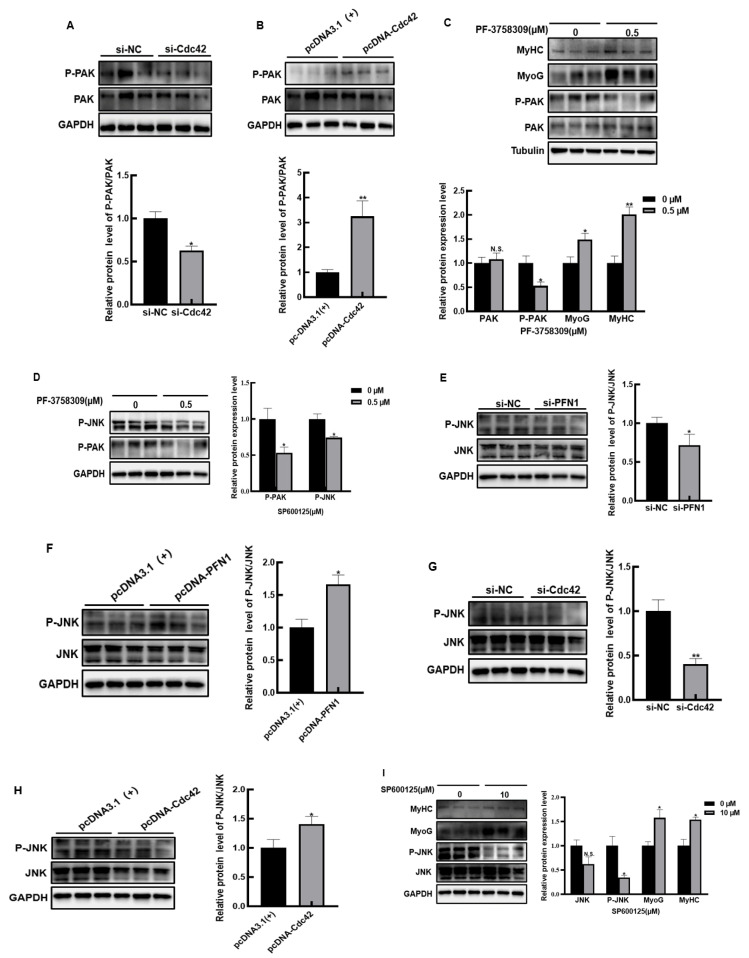
PFN1-Cdc42 negatively regulates bovine skeletal muscle via PAK/JNK signaling pathway: (**A**) Western blot testing and quantitative analysis of the effect of Cdc42 on PAK phosphorylation in si-Cdc42 or si-NC transfected DM2 myoblasts; (**B**) Western blot testing and quantitative analysis of the effect of Cdc42 on PAK phosphorylation in pcDNA-Cdc42 or pcDNA3.1(+) transfected DM2 myoblasts; (**C**) changes analysis of differentiation marker proteins of the DM2 bovine myoblasts with P-PAK inhibitor by Western blot; (**D**) protein level analysis of P-JNK after P-PAK inhibition in DM2 bovine myoblasts by Western blot; (**E**) Western blot detecting and quantitative analysis of the effect of PFN1 on the JNK signal pathway in si-PFN1 or si-NC transfected DM2 myoblasts; (**F**) Western blot detecting and quantitative analysis of the effect of PFN1 on the JNK signal pathway in pcDNA-PFN1 or pcDNA3.1(+) transfected DM2 myoblasts; (**G**) Western blot detecting and quantitative analysis of the effect of Cdc42 on the JNK signal pathway in si-Cdc42 or si-NC transfected DM2 myoblasts; (**H**) the effect analysis of Cdc42 on the JNK signal pathway in pcDNA-Cdc42 or pcDNA3.1(+) transfected DM2 myoblasts; (**I**) change analysis of differentiation marker proteins of the DM2 bovine myoblasts with JNK inhibitor. si-NC stands for negative control, and pcDNA3.1(+) stands for empty vector control negative control. N.S.: non significance, * *p* < 0.05, ** *p* < 0.01.

## Data Availability

The date presented in this study are available on request from the corresponding author upon reasonable request.

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
