# Peer review of "PFN1 Inhibits Myogenesis of Bovine Myoblast Cells via Cdc42-PAK/JNK"

_cells, 2022, doi:10.3390/cells11203188_

Round 1

Reviewer 1 Report

Reviewer comments
The work of Zi et al is a rigorous and progressive analysis on the involvement of PFN1 in the inhibition of the myogenesis of bovine skeletal muscle satellite cells. Showing that PFN1 interacts with Cdc42, the authors convincingly address the function of PFN1 in a mechanistic view : PFN1 interacts with Cdc42 to stimulate Cdc42 activity, thereby activating PAK and JNK signaling pathways to inhibit bovine myoblastic differentiation.
This demonstration is interesting given the involvement of PFN1 in some neurodegenerative diseases such as ALS
The text must however be slightly modified to allow a better reading :
1/ Abstract: define PFN1
2/ Line 9: replace myofifibres by myofibres
3/ In the legend of figures, explain si-NC
4/Line 298: define HCC cells.

Author Response

We have replied point-to-point to all the questions raised by the reviewers. We modified the text of the manuscript to make it easier to read, and we re-write the repeated words. We have added the primer sequence to the materials and methods.

Reviewer 2 Report

In the current study the authors evaluated mechanisms involved in the negative regulation of myoblast differentiation by PFN1. They showed that PFN1 works via Cdc42, Pak, and JNK. At the beginning of the study the authors confirmed that PFN1 is a negative regulator of myoblast differentiation by knocking down or overexpressing PFN1 in cultured bovine myoblasts. Then they identified PFN1 binding partners and showed their relationship with actin and small GTPase. Small GTPase Cdc42 was selected for further evaluation.

Overall, the study is well done, and it presented comprehensive evidence on PFN1 as negative regulation of myoblast differentiation.

English of the manuscript requires significant improvement. Text of the entire manuscript should be carefully evaluated and re-written.

For example:

Lines 30-32 “We want to further know regulatory factors regulating bovine skeletal muscle satellite cell differentiation through experiments, to provide a new regulatory network for beef cattle development”.

Lines 30-32 “In embryonic mice”. In mouse embryos?

Line 303 “we firmly confirmed”

Questions/ suggestions/ limitations of the study.

Title:

“PFN1 inhibits myogenesis of bovine skeletal muscle satellite cells…”

The entire study was done on cultured bovine myoblasts. Myoblasts are very different from the satellite cells in vivo. Please re-write the title and the text of the manuscript reflecting that the entire study was done on the isolated myoblasts in culture and not on bovine satellite cells.

Methods:

Please provide catalog numbers and dilutions used for all antibodies.

Justify the use of GAPDH mRNA as an endogenous control. Was it compared to other housekeeping genes?

Please include Supplementary Table 1 into the manuscript. The reader should not go to the supplementary data to view the primer composition. Table 1 is missing from the review materials in the current submission.

Lines 125-126: All the primer sequences of gene cloning were listed in Supplementary Table (). Table number is missing as well as Table itself from the review materials.

Results

Figures S1, S2 and S3 are missing from the current submission. I was unable to review them. Video S1 is missing from the current submission.

Discussion

The last paragraph before the Conclusions is too long. It can be divided into two-three separate paragraphs.

Please re-write the text to clarify logical connections between sentences.

Conclusions

Please replace skeletal muscle satellite cells with myoblasts. Satellite cells were not evaluated in this study. Both quiescent and activated satellite cells in vivo are very different from cultured myoblasts evaluated in this study.

Author Response

(The authors gave the same response as above.)
